# Evaluation of oral dextrose gel for prevention of neonatal hypoglycemia (hPOD): A multicenter, double-blind randomized controlled trial

Jane E. Harding[1]*, Joanne E. Hegarty[1,2], Caroline A. Crowther[1], Richard P. Edlin[1], Gregory D. Gamble[1], Jane M. Alsweiler[2,3], for the hPOD Study Group¶

1 Liggins Institute, University of Auckland, Auckland, New Zealand, 2 Newborn Services, Auckland City Hospital, Auckland, New Zealand, 3 Department of Paediatrics: Child and Youth Health, University of Auckland, Auckland, New Zealand

¶ Membership of the hPOD Study Group is provided in the Acknowledgments.
* j.harding@auckland.ac.nz

**Data Availability Statement:** Data and associated documentation are available to other users under

## Abstract

### Background

Neonatal hypoglycemia is common and can cause brain injury. Buccal dextrose gel is effective for treatment of neonatal hypoglycemia, and when used for prevention may reduce the incidence of hypoglycemia in babies at risk, but its clinical utility remains uncertain.

### Methods and findings

We conducted a multicenter, double-blinded, placebo-controlled randomized trial in 18 New Zealand and Australian maternity hospitals from January 2015 to May 2019. Babies at risk of neonatal hypoglycemia (maternal diabetes, late preterm, or high or low birthweight) without indications for neonatal intensive care unit (NICU) admission were randomized to 0.5 ml/kg buccal 40% dextrose or placebo gel at 1 hour of age. Primary outcome was NICU admission, with power to detect a 4% absolute reduction. Secondary outcomes included hypoglycemia, NICU admission for hypoglycemia, hyperglycemia, breastfeeding at discharge, formula feeding at 6 weeks, and maternal satisfaction. Families and clinical and study staff were unaware of treatment allocation. A total of 2,149 babies were randomized (48.7% girls). NICU admission occurred for 111/1,070 (10.4%) randomized to dextrose gel and 100/1,063 (9.4%) randomized to placebo (adjusted relative risk [aRR] 1.10; 95% CI 0.86, 1.42; *p* = 0.44). Babies randomized to dextrose gel were less likely to become hypoglycemic (blood glucose < 2.6 mmol/l) (399/1,070, 37%, versus 448/1,063, 42%; aRR 0.88; 95% CI 0.80, 0.98; *p* = 0.02) although NICU admission for hypoglycemia was similar between groups (65/1,070, 6.1%, versus 48/1,063, 4.5%; aRR 1.35; 95% CI 0.94, 1.94; *p* = 0.10). There were no differences between groups in breastfeeding at discharge from hospital (aRR 1.00; 95% CI 0.99, 1.02; *p* = 0.67), receipt of formula before discharge (aRR 0.99; 95% CI 0.92, 1.08; *p* = 0.90), and formula feeding at 6 weeks (aRR 1.01; 95% CI 0.93, 1.10;

the data sharing arrangements provided by the Maternal and Perinatal Research Hub, based at the Liggins Institute, University of Auckland (https://wiki.auckland.ac.nz/researchhub).The data dictionary and metadata are published on the University of Auckland's data repository Figshare, which allocates a DOI and thus makes these details searchable and available indefinitely. Researchers are able to use this information and the provided contact address (researchhub@auckland.ac.nz) to request a de-identified dataset through the Data Access Committee of the Liggins Institute. Data will be shared with researchers who provide a methodologically sound proposal and have appropriate ethical approval, where necessary, to achieve the research aims in the approved proposal. Data requestors are required to sign a Data Access Agreement that includes a commitment to using the data only for the specified proposal, not to attempt to identify any individual participant, a commitment to secure storage and use of the data, and to destroy or return the data after completion of the project. The Liggins Institute reserves the right to charge a fee to cover the costs of making data available, if needed, for data requests that require additional work to prepare.

**Funding:** This trial was funded by a grant from the Health Research Council of New Zealand (13/131) to JEHa, including partial salary support for JEHa, JEHe, RPE, GDG, JMA. The funder had no role in study design, data collection and analysis, decision to publish, or preparation of the manuscript.

**Competing interests:** I have read the journal's policy and the authors of this manuscript have the following competing interests: JH has in the past received an unrestricted research grant from Biomed Auckland, who manufacture dextrose gel. That sponsor had no role in this study, and in particular, no role in study design, data collection and analysis, decision to publish, or preparation of the manuscript.

**Abbreviations:** aMD, adjusted mean difference; aRR, adjusted relative risk; NICU, neonatal intensive care unit.

$p$ = 0.81), and there was no hyperglycemia. Most mothers (95%) would recommend the study to friends. No adverse effects, including 2 deaths in each group, were attributable to dextrose gel. Limitations of this study included that most participants (81%) were infants of mothers with diabetes, which may limit generalizability, and a less reliable analyzer was used in 16.5% of glucose measurements.

## Conclusions

In this placebo-controlled randomized trial, prophylactic dextrose gel 200 mg/kg did not reduce NICU admission in babies at risk of hypoglycemia but did reduce hypoglycemia. Long-term follow-up is needed to determine the clinical utility of this strategy.

## Trial registration

ACTRN 12614001263684.

## Author summary

### Why was this study done?

- Hypoglycemia (low blood glucose level) is common in newborn babies and can cause brain injury, even if it is transient and treated.

- Dextrose (sugar) gel rubbed inside the baby's cheek is widely used to treat hypoglycemia, and is noninvasive, inexpensive, and safe.

- One study previously has shown that dextrose gel can be used as a preventative to reduce the incidence of hypoglycemia, but it is not known if this improves clinically important outcomes like admission to newborn intensive care.

### What did the researchers do and find?

- We recruited from 18 centers in New Zealand and Australia 2,149 babies who were born at risk of neonatal hypoglycemia but who were not likely to need intensive care for other reasons.

- Babies were allocated at random to receive a single dose of dextrose gel or placebo gel at 1 hour after birth, and had blood glucose levels measured at 2 hours, followed by routine care.

- Preventative dextrose gel did not decrease admission to newborn intensive care but did decrease the incidence of hypoglycemia (secondary outcome), with 21 babies needing to be treated to prevent 1 case of hypoglycemia.

- There were no effects on breastfeeding, no high blood glucose levels, and no other adverse effects.

### What do these findings mean?

- Clinicians and clinical guideline groups should consider whether needing to treat 21 babies to prevent 1 case of hypoglycemia with no reduction in neonatal intensive care admission warrants introduction of this prevention strategy into practice at this time.

- Since the main reason for preventing hypoglycemia is to prevent brain injury, it will be important to assess the effect of this prevention strategy on the later development of children in this cohort.

## Introduction

Neonatal hypoglycemia is common, affecting up to 15% of newborn babies [1] and 50% of those with risk factors (preterm, infant of a mother with diabetes, or high or low birthweight) [2,3]. First-line treatment with oral 40% dextrose gel in addition to feeding is safe and effective [3], but if hypoglycemia persists, intravenous dextrose is recommended [4]. This commonly requires admission to the neonatal intensive care unit (NICU), separating mother and baby and disrupting the establishment of breastfeeding.

Hypoglycemia can cause brain damage and death, and babies born at risk have an increased risk of developmental delay in later life [5]. Even transient and treated hypoglycemia has been associated with impaired visual-motor coordination and executive function at 4.5 years [6], and with poorer performance on standard school testing of literacy and mathematics at 10 years [7]. This suggests that effective treatment may not be sufficient to avoid brain injury and that prevention of neonatal hypoglycemia would be desirable. However, there are currently no strategies, beyond early feeding, for prevention of neonatal hypoglycemia.

We have previously shown in a dose-finding study that 40% dextrose gel given prophylactically to babies at risk reduces the incidence of neonatal hypoglycemia [8]. We therefore undertook this multicenter randomized trial to assess whether prophylactic dextrose gel given to babies at risk of neonatal hypoglycemia reduces admission to NICU.

## Methods

### Design

This multicenter, double-blinded, 2-arm, parallel, placebo-controlled randomized trial was conducted at 18 Australian and New Zealand maternity hospitals (trial registration ACTRN12614001263684). The study protocol has been published previously [9]. Babies were eligible if they were born at risk of hypoglycemia (defined as at least 1 of the following: preterm [<37 weeks' gestation], infant of a mother with diabetes [any type], small [birthweight < 2.5 kg or <10th centile on population or customized birthweight chart], or large [birthweight > 4.5 kg or >90th centile on population or customized birthweight chart]) and also satisfied all of the following: ≥35 weeks' gestation; birthweight ≥ 2.2 kg; <1 hour old; no apparent indication for NICU admission; unlikely to require NICU admission for any other reason, e.g., respiratory distress; and mother intended to breastfeed. Babies were not eligible if they had a major congenital abnormality, had received formula feed or intravenous fluids, or had been admitted to NICU, or admission to NICU was imminent.

### Ethics statement

This study was approved by the New Zealand Health and Disability Ethics Committee (13NTA8), the Human Research Ethics Committee at the Women's and Children's Hospital, Adelaide (HREC/16/WCHN/86), and the institutional review committees at each participating hospital. Parents gave written informed consent, which was sought before birth whenever possible.

### Randomization and masking

The randomization schedule was prepared by the study statistician, who was not involved with any clinical aspect of the trial, and was stratified by study site and reason for risk of hypoglycemia (infant mother with diabetes, preterm, small, or large) with varied block size using the Plan procedure of SAS (version 9.4; SAS Institute, Cary, NC, US). Staff at the study sites accessed a centralized internet-based randomization service within the first hour after the birth to receive a study number that corresponded to a study treatment pack containing a single pre-packaged syringe of 40% dextrose gel or identical-appearing 2% hydroxymethylcellulose placebo gel (1:1 ratio). Families, study and site staff, and investigators were all blinded to treatment allocation.

### Procedures

Randomized babies received a single dose of 0.5 ml/kg study gel at 1 hour after birth. This dose (200 mg/kg of 40% dextrose) was selected based on the pre-hPOD dosage trial [8] as having greatest efficacy with fewest limitations. The buccal mucosa was dried with a gauze swab before the study gel was massaged into the mucosa, followed by a breast feed. Blood glucose concentration was measured at 2 hours of age, and then according to hospital standard practice for monitoring babies at risk of hypoglycemia. This usually included pre-feed blood glucose concentration measurements 2–4 hourly for at least the first 12 hours, and until there had been 3 consecutive measurements of blood glucose $\geq 2.6$ mmol/l. The study protocol specified that all blood glucose concentrations should be analyzed using a glucose oxidase method, either with a portable blood glucose analyzer (e.g., iSTAT, Abbott Laboratories, Abbott Park, IL, US) or a combined metabolite/blood gas analyzer (e.g., ABL 700, Radiometer, Copenhagen, Denmark). Babies who became hypoglycemic were treated according to standard hospital clinical practice, which in most cases was initially supplementary feeds and then treatment with 40% dextrose gel, followed by intravenous dextrose if required.

Parents of included babies were contacted on day 3 (by telephone if already discharged home) to complete a questionnaire about current feeding, and at 6 weeks to complete a questionnaire about current feeding, parental satisfaction with participation in the trial, and health status of the baby.

### Outcomes

The primary outcome was admission to NICU (or Special Care Baby Unit for hospitals that used that name) for >4 hours. Secondary outcomes were hypoglycemia (any blood glucose concentration < 2.6 mmol/l in the first 48 hours), admission to NICU for hypoglycemia, hyperglycemia (any blood glucose concentration > 10 mmol/l), full or exclusive breastfeeding at discharge from hospital, receipt of any formula before discharge from hospital, formula feeding at 6 weeks of age, maternal satisfaction with study participation; cost of care until primary discharge home (to be reported separately), and neurosensory disability at 2 years' corrected age (follow-up in progress). Adverse events were monitored by an independent safety

monitoring committee, and were defined as seizures (serious adverse effect), death (serious adverse effect), hyperglycemia (defined as above), late hypoglycemia (blood glucose concentration < 2.6 mmol/l for the first time after 12 hours of age), delayed feeding (failure to establish breastfeeding without supplements by the end of day 3), and systemic sepsis.

## Sample size

Based on our previous data from Auckland City [8] and Waikato Hospitals [3], we estimated that 10% of at-risk babies would require admission to NICU. A trial of 2,129 babies (1,014 in each arm, with continuity correction and allowing for a 5% dropout rate) would have 90% power to detect a 40% relative reduction (absolute reduction of 4%) in admission to NICU from 10% to 6% with a 2-sided alpha of 0.05.

## Statistical analysis

The trial was overseen by an independent data monitoring committee and safety monitoring committee. No interim analyses were planned or undertaken. All analyses were prespecified and carried out using a modified intention-to-treat approach, in which babies randomized in error (i.e., who did not meet eligibility criteria at randomization) were excluded, but all other babies were analyzed in the groups to which they were allocated. Babies for whom the primary outcome was not available were assumed to have been admitted to NICU (conservative analysis), but there was no other imputation for missing data.

Between-group differences in binary outcomes (admission to NICU, hypoglycemia) were analyzed using mixed-effects general linear models assuming a binary distribution and log link function to obtain robust estimates of relative risk with 95% confidence intervals after prespecified adjustment for randomization stratification variables: study site and prioritized primary reason for risk of hypoglycemia (infant of a mother with diabetes, preterm, small, large) as fixed effects, and maternal unique identifier as a random effect clustering term to account for the non-independence of multiple births. For continuous outcomes the same models were fitted but a Gaussian distribution was assumed and the identity link function was used to obtain mean differences with 95% confidence intervals.

In exploratory analyses the following terms were included: study site, maternal unique identifier as a clustering term, and (a) infant of a mother with diabetes and gestational age and birthweight $z$-score, (b) infant of a mother with diabetes and preterm (<37 weeks' gestation) and birthweight $z$-score, (c) sex and mode of birth (vaginal or cesarean section), or (d) open label treatment with 40% dextrose gel. All analyses were prespecified unless otherwise stated.

To explore changes in blood glucose concentration over time, a mixed-models approach to repeated measures was used. Time was rounded into hour bins, and time, treatment, and their interaction effects were fitted to a model that included the randomization stratification variables. Significant interaction effects were further explored by between-treatment-group comparison of the adjusted marginal means at each hour with false discovery rate $p$-values.

The primary outcome was tested at the 5% significance level. No adjustment to the critical significance level was made for any secondary, sensitivity, or exploratory analyses other than for the changes in glucose over time described above. All analyses were conducted using SAS (version 9.4, SAS Institute, Cary, NC, US).

## Results

Eighteen participating hospitals recruited 2,149 babies between 9 January 2015 and 5 May 2019 (range 5–535 babies per site). Sixteen babies were randomized in error and were excluded from the analysis, leaving 2,133 in the intention-to-treat analysis, 1,070 randomized to

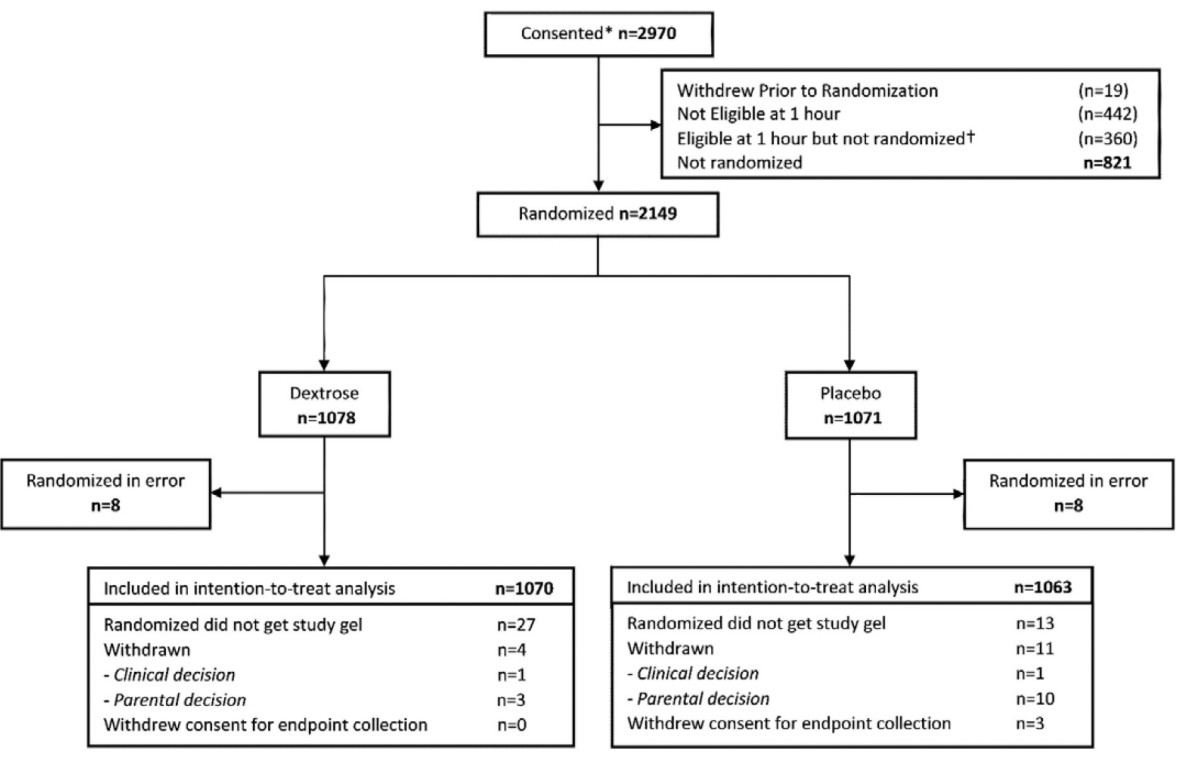

**Fig 1. Participant flowchart.**

dextrose gel and 1,063 to placebo (Fig 1). This includes 40 babies who were randomized but did not receive study gel, and 15 who withdrew after randomization. Overall mean (SD) birthweight was 3,321 (603) g (infants of mothers with diabetes, 3,385 [503] g; preterm, 2,726 [328] g; small, 2,532 [217] g; large, 4,458 [419] g). Groups were well balanced for maternal and baby demographic variables (Table 1). The most common reason for risk of hypoglycemia was being an infant of a mother with diabetes (81% of each group), and 18% had more than 1 risk factor.

## Study conduct

Most babies (2,093/2,133; 98%) received the allocated study gel. Study gel was well tolerated by 2,044/2,097 (97%) babies (defined as none or only a few drops of gel spilled). Most blood glucose measurements were done using a glucose oxidase method (9,583/11,481; 83.5%), and the mean (SD) number of glucose measurements per baby was 7.8 (4.0) in those who became hypoglycemic and 3.8 (1.5) in those who did not, with no differences between treatment groups. At 6 weeks, 69% of mothers (270/389) in the dextrose gel group correctly guessed their baby's study group, compared with 44% of mothers (144/331) in the placebo gel group ($p < 0.001$).

## Primary outcome

Three families withdrew consent to collect the primary outcome of NICU admission (all in the placebo group) and were therefore assumed to have been admitted to NICU for the intention-to-treat analysis. The overall rate of NICU admission was 9.9%, and was similar in babies

**Table 1. Characteristics of mothers and babies randomized to placebo or dextrose gel.**

| Characteristic | Placebo | Dextrose |
|---|---|---|
| Mothers (N = 2,051) | N = 1,025 | N = 1,026 |
| Maternal age (years) | 32.2 (5.4) | 32.2 (5.3) |
| *Prioritized ethnicity* | | |
| Aboriginal/Torres Strait Islander | 24 (2.3%) | 9 (0.9%) |
| Maori | 122 (11.9%) | 116 (11.3%) |
| Pacific | 56 (5.5%) | 60 (5.9%) |
| Asian | 346 (33.8%) | 351 (34.2%) |
| Indian | 162 (15.8%) | 166 (16.2%) |
| Other | 76 (7.4%) | 76 (7.4%) |
| European | 239 (23.3%) | 248 (24.2%) |
| *Diabetes* | | |
| Type 1 diabetes | 31 (3.7%) | 38 (4.6%) |
| Type 2 diabetes | 66 (8.0%) | 57 (6.8%) |
| Gestational diabetes | 732 (88.3%) | 740 (88.6%) |
| *Diabetes management** | | |
| Diet | 407 (39.8%) | 417 (40.7%) |
| Metformin | 258 (25.2%) | 247 (24.1%) |
| Insulin | 451 (44.1%) | 432 (42.2%) |
| Antenatal corticosteroids | 41 (4.0%) | 46 (4.5%) |
| Prelabor prolonged rupture of membranes* | 81 (7.9%) | 87 (8.5%) |
| Chorioamnionitis* | 5 (0.5%) | 5 (0.5%) |
| *Mode of delivery* | | |
| Normal vaginal | 494 (48.2%) | 464 (45.2%) |
| Instrumental vaginal | 123 (12.0%) | 125 (12.2%) |
| Cesarean section | 405 (39.5%) | 436 (42.5%) |
| Babies (N = 2,133) | N = 1,063 | N = 1,070 |
| Singleton | 999 (94.3%) | 991 (92.7%) |
| Girls | 523 (49.2%) | 515 (48.1%) |
| Gestational age (weeks) | 38.5 (1.1) | 38.4 (1.1) |
| Birthweight (g) | 3,313 (594) | 3,328 (613) |
| Length (cm) | 49.9 (2.5) | 50.0 (2.8) |
| Head circumference (cm) | 34.5 (1.7) | 34.6 (1.8) |
| Birthweight $z$-score | 0.20 (1.13) | 0.25 (1.13) |
| Length $z$-score | 0.23 (0.98) | 0.30 (1.09) |
| Head circumference $z$-score | 0.35 (1.09) | 0.39 (1.16) |
| Apgar score < 7 at 5 minutes[#] | 5 (0.5%) | 10 (0.9%) |
| *Primary reason for risk of hypoglycemia* | | |
| Infant of mother with diabetes | 856 (80.5%) | 863 (80.7%) |
| Preterm (<37 weeks' gestation) | 76 (7.2%) | 75 (7.0%) |
| Small (<2.5 kg or <10th centile) | 84 (7.9%) | 83 (7.8%) |
| Large (>4.5 kg or >90th centile) | 47 (4.4%) | 49 (4.6%) |
| Two risk factors | 178 (16.8%) | 175 (16.5%) |
| Three risk factors | 7 (0.7%) | 14 (1.3%) |

Data are $n$ (%) or mean (SD).

*$n$ = 1,022 placebo, 1,024 dextrose.

[#]$n$ = 1,060 placebo, 1,069 dextrose.

**Table 2. Primary and key secondary outcomes and potential adverse effects.**

| Outcome | Placebo N = 1,063 | Dextrose N = 1,070 | aRR or aMD | 95% CI | p-Value |
|---|---|---|---|---|---|
| Admission to NICU | 100/1,063 (9.4%) | 111/1,070 (10.4%) | 1.10 | 0.86, 1.42 | 0.44 |
| Hypoglycemia | 448/1,063 (42.1%) | 399/1,070 (37.3%) | 0.88 | 0.80, 0.98 | 0.02 |
| NICU admission for hypoglycemia | 48/1,063 (4.5%) | 65/1,070 (6.1%) | 1.35 | 0.94, 1.94 | 0.10 |
| Treated for hypoglycemia | 337/1,063 (31.7%) | 307/1,070 (28.7%) | 0.90 | 0.79, 1.02 | 0.09 |
| Treated with open label dextrose gel | 325/1,063 (30.6%) | 299/1,070 (27.9%) | 0.90 | 0.80, 1.03 | 0.12 |
| Recurrent hypoglycemia | 142/1,063 (13.4%) | 131/1,070 (12.2%) | 0.91 | 0.73, 1.14 | 0.43 |
| Severe hypoglycemia | 105/1,063 (9.9%) | 99/1,070 (9.3%) | 0.93 | 0.72, 1.20 | 0.58 |
| Late hypoglycemia* | 109/606 (18.0%) | 104/601 (17.3%) | 0.97 | 0.76, 1.24 | 0.83 |
| First blood glucose concentration (mmol/l)# | 2.97 (0.69) (n = 1,049) | 3.16 (0.77) (n = 1,059) | 0.19 | 0.13, 0.25 | <0.001 |
| Breastfeeding at hospital discharge | 1,010/1,053 (95.9%) | 1,027/1,063 (96.6%) | 1.00 | 0.99, 1.02 | 0.67 |
| Received formula prior to discharge | 512/1,053 (48.6%) | 509/1,065 (47.8%) | 0.99 | 0.92, 1.08 | 0.90 |
| Delayed breastfeeding | 388/1,027 (37.8%) | 393/1,041 (37.8%) | 1.01 | 0.91, 1.12 | 0.86 |
| Formula feeding at 6 weeks | 473/957 (49.4%) | 481/981 (49.0%) | 1.01 | 0.93, 1.10 | 0.81 |
| Would take part again | 882/951 (92.7%) | 926/973 (95.2%) | 1.03 | 1.00, 1.05 | 0.03 |
| Would recommend study to friends | 901/951 (94.7%) | 929/974 (95.4%) | 1.01 | 0.99, 1.03 | 0.54 |

Data are n/N (%) or mean (SD) (n). Adjustments are for multiple births, study site, and primary reason for risk of hypoglycemia. Hypoglycemia is blood glucose concentration < 2.6 mmol/l; severe hypoglycemia is blood glucose concentration < 2.0 mmol/l.

*Blood glucose < 2.6 mmol/l for the first time after 12 hours of age.

#Measured post-randomization, 1–4 hours after birth.

aMD, adjusted mean difference; aRR, adjusted relative risk; CI, confidence interval; NICU, neonatal intensive care unit.

randomized to dextrose and placebo gel (Table 2). The mean (SD) age of NICU admission was 11.0 (12.2) hours in babies admitted for hypoglycemia and 22.1 (32.5) hours in babies admitted for other reasons, with no difference between treatment groups.

## Secondary outcomes

Babies randomized to dextrose gel had higher initial blood glucose concentrations (mean difference 0.19 mmol/l; 95% CI 0.13, 0.25 mmol/l; $p < 0.001$) and were less likely to become hypoglycemic (adjusted relative risk [aRR] 0.88; 95% CI 0.80, 0.98; $p = 0.02$) (Table 2). However, the rate of treatment for hypoglycemia did not differ between groups (aRR 0.90; 95% CI 0.79, 1.02; $p = 0.09$), nor did the rate of NICU admission for hypoglycemia (aRR 1.35; 95% CI 0.94, 1.94; $p = 0.10$). Overall, 30% of babies received treatment for hypoglycemia with open label dextrose gel (644/2,133) and 3.4% received intravenous dextrose (72/2,133), with no differences between treatment groups. There were no differences between treatment groups in delayed breastfeeding, breastfeeding at discharge from hospital, receipt of formula before discharge, or formula feeding at 6 weeks of age (Table 2). No babies became hyperglycemic (blood glucose > 10 mmol/l). Maternal satisfaction with the study at 6 weeks was high, with 95% of each group reporting that they would recommend the study to their friends; slightly more mothers in the dextrose gel group reported that they would take part in the study again (926/973, 95%, versus 882/951, 93%; aRR 1.03; 95% CI 1.00, 1.05; $p = 0.03$).

## Adverse effects

Two babies in each group died before discharge home; no deaths were considered likely to be related to the study intervention. One baby randomized to placebo gel had seizures 3 days

after discharge, which were thought to be benign. Sepsis was suspected in 17 babies in each group (aRR 0.99; 95% CI 0.52, 1.93; $p$ = 0.99), but confirmed in only 1 baby in the dextrose group. No baby had hyperglycemia (blood glucose > 10 mmol/l). Late hypoglycemia occurred in 213/1,207 (17.6%) babies with glucose measurements after 12 hours of age and was similar in both treatment groups. Breastfeeding was delayed in 38% of each group, and there were no differences between groups in the rate of full or exclusive breastfeeding, formula feeding before hospital discharge, or formula feeding at 6 weeks of age (Table 2).

## Exploratory analyses

**Subgroup analyses.** The rate of NICU admission varied widely across study sites (range 6.7% to 32.1%), and was higher in Australian than in New Zealand centers (113/774, 14.6%, versus 98/1,359, 7.2%; aRR 2.12; 95% CI 1.64, 2.75; $p$ < 0.001). However, there was no evidence that dextrose gel altered the rate of NICU admission compared to placebo gel in different countries, in level 3 versus level 2 centers, or in the 4 centers that together recruited 78% of the babies (Fig 2A). There was also no evidence that the effect of dextrose compared to placebo gel was different in babies with different risk factors for hypoglycemia or different modes of birth, or for boys compared to girls (Fig 2B).

Secondary outcomes were also not affected by study site, primary risk factor for hypoglycemia, or infant sex. However, the rate of hypoglycemia was lower in the dextrose gel group than in the placebo group in babies born vaginally (aRR 0.81; 95% CI 0.70, 0.94; $p$ < 0.01) but not in those born by cesarean section (aRR 0.97; 95% CI 0.83, 1.12; $p$ = 0.65) (Table 3). In post hoc analysis, the initial blood glucose concentration was higher in the dextrose gel group than the placebo group in babies born vaginally (mean [SD] 3.3 [0.8] mmol/l, $n$ = 604, versus 3.0 [0.7] mmol/l, $n$ = 625; adjusted mean difference [aMD] 0.27; 95% CI 0.19, 0.35; $p$ < 0.001) but not in those born by cesarean section (mean [SD] 3.0 [0.8] mmol/l, $n$ = 455, versus 2.9 [0.7] mmol/l, $n$ = 424; aMD 0.08; 95% CI −0.02, 0.18 mmol/l; $p$ = 0.11).

**Sensitivity analyses.** Sensitivity analyses excluding babies with protocol deviations, babies who did not receive any of the assigned study gel (modified per protocol analysis), or babies for whom the primary outcome was not known did not change any of the findings (Table 3). Findings were also similar if only glucose measurements using a glucose oxidase method were included (Table 3).

**Other exploratory analyses.** Adjustment for other potential confounders (see "Statistical analysis") did not change any of the key findings, with relative risks for NICU admission of 1.08–1.12 ($p$ = 0.57–0.37) across the 5 prespecified adjustments. There was no evidence that the effect of dextrose gel was related to the rate of NICU admission, or to the rate of hypoglycemia, in individual centers (Fig 3).

Blood glucose concentration increased in both groups over the first day (time $p$ < 0.001), and was higher in the dextrose gel group than the placebo group (treatment × time $p$ < 0.001) specifically at 2 hours of age ($p$ < 0.01; Fig 4).

## Discussion

We have previously shown that a single dose of 200 mg/kg prophylactic dextrose gel reduced the incidence of hypoglycemia in babies at risk [8]. We therefore hypothesized that in at-risk but otherwise well babies, prophylactic dextrose gel may reduce NICU admission, with potential health, societal, and cost benefits. However, in this large multicenter randomized trial, dextrose gel prophylaxis did not reduce NICU admission. There are several possible reasons for this. First, our inclusion criteria were intended to exclude babies requiring early NICU admission for reasons other than hypoglycemia, and the relatively older age of those who were

## A. NICU Admission

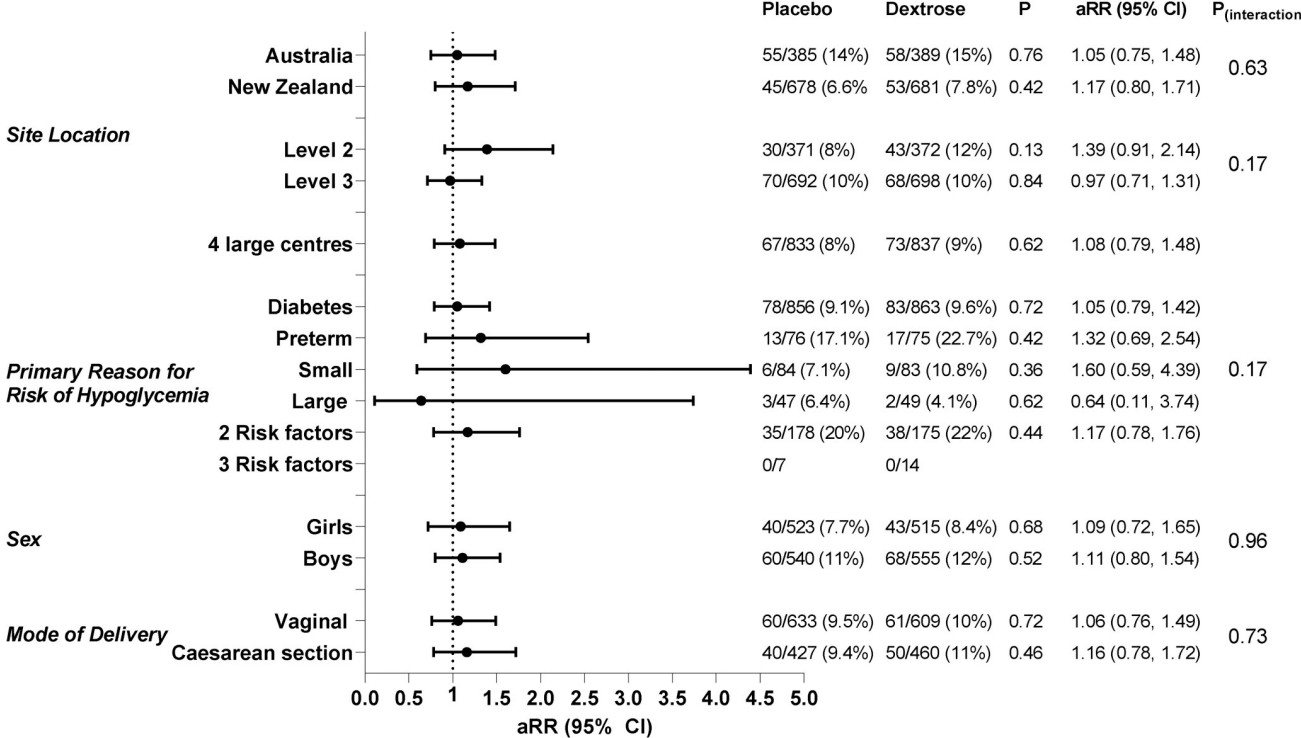

| | Placebo | Dextrose | P | aRR (95% CI) | P(interaction) |
|---|---|---|---|---|---|
| **Site Location** | | | | | |
| Australia | 55/385 (14%) | 58/389 (15%) | 0.76 | 1.05 (0.75, 1.48) | 0.63 |
| New Zealand | 45/678 (6.6%) | 53/681 (7.8%) | 0.42 | 1.17 (0.80, 1.71) | |
| Level 2 | 30/371 (8%) | 43/372 (12%) | 0.13 | 1.39 (0.91, 2.14) | 0.17 |
| Level 3 | 70/692 (10%) | 68/698 (10%) | 0.84 | 0.97 (0.71, 1.31) | |
| 4 large centres | 67/833 (8%) | 73/837 (9%) | 0.62 | 1.08 (0.79, 1.48) | |
| **Primary Reason for Risk of Hypoglycemia** | | | | | |
| Diabetes | 78/856 (9.1%) | 83/863 (9.6%) | 0.72 | 1.05 (0.79, 1.42) | |
| Preterm | 13/76 (17.1%) | 17/75 (22.7%) | 0.42 | 1.32 (0.69, 2.54) | |
| Small | 6/84 (7.1%) | 9/83 (10.8%) | 0.36 | 1.60 (0.59, 4.39) | 0.17 |
| Large | 3/47 (6.4%) | 2/49 (4.1%) | 0.62 | 0.64 (0.11, 3.74) | |
| 2 Risk factors | 35/178 (20%) | 38/175 (22%) | 0.44 | 1.17 (0.78, 1.76) | |
| 3 Risk factors | 0/7 | 0/14 | | | |
| **Sex** | | | | | |
| Girls | 40/523 (7.7%) | 43/515 (8.4%) | 0.68 | 1.09 (0.72, 1.65) | 0.96 |
| Boys | 60/540 (11%) | 68/555 (12%) | 0.52 | 1.11 (0.80, 1.54) | |
| **Mode of Delivery** | | | | | |
| Vaginal | 60/633 (9.5%) | 61/609 (10%) | 0.72 | 1.06 (0.76, 1.49) | 0.73 |
| Caesarean section | 40/427 (9.4%) | 50/460 (11%) | 0.46 | 1.16 (0.78, 1.72) | |

## B. Hypoglycemia

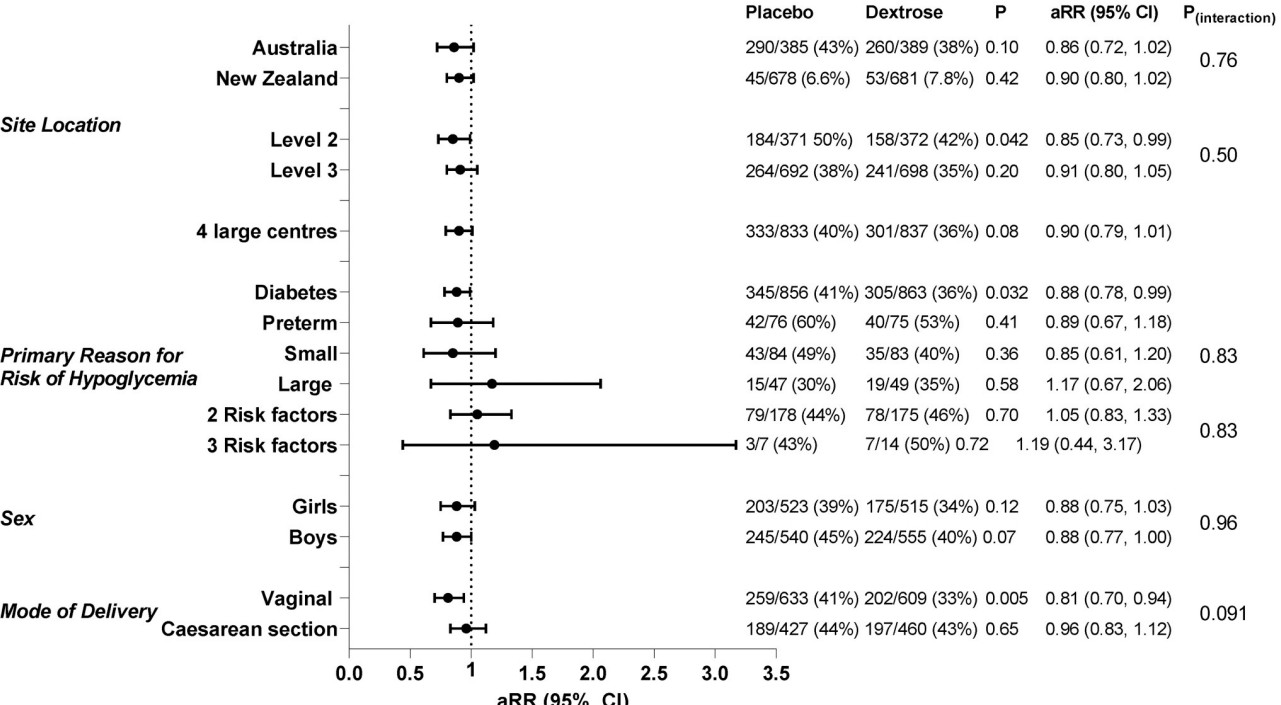

| | Placebo | Dextrose | P | aRR (95% CI) | P(interaction) |
|---|---|---|---|---|---|
| **Site Location** | | | | | |
| Australia | 290/385 (43%) | 260/389 (38%) | 0.10 | 0.86 (0.72, 1.02) | 0.76 |
| New Zealand | 45/678 (6.6%) | 53/681 (7.8%) | 0.42 | 0.90 (0.80, 1.02) | |
| Level 2 | 184/371 50%) | 158/372 (42%) | 0.042 | 0.85 (0.73, 0.99) | 0.50 |
| Level 3 | 264/692 (38%) | 241/698 (35%) | 0.20 | 0.91 (0.80, 1.05) | |
| 4 large centres | 333/833 (40%) | 301/837 (36%) | 0.08 | 0.90 (0.79, 1.01) | |
| **Primary Reason for Risk of Hypoglycemia** | | | | | |
| Diabetes | 345/856 (41%) | 305/863 (36%) | 0.032 | 0.88 (0.78, 0.99) | |
| Preterm | 42/76 (60%) | 40/75 (53%) | 0.41 | 0.89 (0.67, 1.18) | |
| Small | 43/84 (49%) | 35/83 (40%) | 0.36 | 0.85 (0.61, 1.20) | 0.83 |
| Large | 15/47 (30%) | 19/49 (35%) | 0.58 | 1.17 (0.67, 2.06) | |
| 2 Risk factors | 79/178 (44%) | 78/175 (46%) | 0.70 | 1.05 (0.83, 1.33) | 0.83 |
| 3 Risk factors | 3/7 (43%) | 7/14 (50%) | 0.72 | 1.19 (0.44, 3.17) | |
| **Sex** | | | | | |
| Girls | 203/523 (39%) | 175/515 (34%) | 0.12 | 0.88 (0.75, 1.03) | 0.96 |
| Boys | 245/540 (45%) | 224/555 (40%) | 0.07 | 0.88 (0.77, 1.00) | |
| **Mode of Delivery** | | | | | |
| Vaginal | 259/633 (41%) | 202/609 (33%) | 0.005 | 0.81 (0.70, 0.94) | 0.091 |
| Caesarean section | 189/427 (44%) | 197/460 (43%) | 0.65 | 0.96 (0.83, 1.12) | |

**Fig 2. Subgroup analyses for the effects of dextrose gel versus placebo on risk of neonatal intensive care unit (NICU) admission and hypoglycemia.** (A) NICU admission; (B) hypoglycemia. Horizontal lines indicate adjusted relative risks (aRRs) and 95% confidence intervals.

**Table 3. Prespecified sensitivity analyses.**

| Outcome | Placebo N = 1,063 | Dextrose N = 1,070 | aRR | 95% CI | p-Value |
|---|---|---|---|---|---|
| *Excluding protocol deviations* | | | | | |
| NICU admission | 91/1,017 (9.0%) | 95/1,013 (9.4%) | 1.06 | 0.80, 1.39 | 0.70 |
| Hypoglycemia | 431/1,017 (42.4%) | 368/1,013 (36.3%) | 0.86 | 0.77, 0.96 | 0.01 |
| *Excluding babies who did not receive assigned study gel* | | | | | |
| NICU admission | 96/1,050 (9.1%) | 99/1,043 (9.5%) | 1.05 | 0.80, 1.37 | 0.74 |
| Hypoglycemia | 445/1,050 (42.4%) | 382/1,043 (36.6%) | 0.87 | 0.78, 0.96 | 0.01 |
| *Including only glucose oxidase measurements* | | | | | |
| Hypoglycemia—overall | 425/1,060 (40.1%) | 378/1,072 (35.3%) | 0.88 | 0.79, 0.98 | 0.02 |
| Hypoglycemia—vaginal births | 245/633 (38.7%) | 189/609 (31.0%) | 0.80 | 0.69, 0.94 | 0.01 |
| Hypoglycemia—cesarean section births | 180/427 (42.2%) | 189/460 (41.1%) | 0.97 | 0.84, 1.14 | 0.74 |
| *Including only glucose oxidase measurements and excluding protocol deviations* | | | | | |
| Hypoglycemia | 412/1,017 (40.5%) | 351/1,013 (34.7%) | 0.86 | 0.77, 0.96 | 0.01 |
| *Including only glucose oxidase measurements and excluding babies who did not receive assigned study gel* | | | | | |
| Hypoglycemia | 423/1,050 (40.3%) | 363/1,043 (34.8%) | 0.86 | 0.78, 0.97 | 0.01 |

Data are *n* (%) or mean (SD). Adjustments are for multiple births, study site, and primary reason for risk of hypoglycemia. Hypoglycemia is blood glucose concentration < 2.6 mmol/l.

aRR, adjusted relative risk; CI, confidence interval; NICU, neonatal intensive care unit.

admitted (mean 22 hours) suggests that this was effective. Nevertheless, 10% were eventually admitted to NICU, only half of these for hypoglycemia. This suggests that it is difficult in the first hour after birth to identify all "otherwise well" babies who may be most likely to benefit if hypoglycemia can be prevented.

Second, the overall incidence of hypoglycemia (40%) was lower than in our previous studies of similar cohorts of babies at risk who were screened according to standard protocols using accurate methods (50%) [2,3]. In this pragmatic multicenter trial, frequency and duration of glucose screening, and thresholds for intervention, were not specified in the trial protocol and varied across study sites. Since detection of hypoglycemia largely depends on how carefully it is sought [10], and many sites used less rigorous screening protocols than previous reports [3,8], it is possible that some hypoglycemia was not detected. However, there was no evidence that the effect of dextrose gel was related to the incidence of hypoglycemia across different study sites, suggesting that this is not likely to have substantially influenced our findings.

Third, this trial confirms the efficacy of prophylactic dextrose gel in reducing the incidence of hypoglycemia in babies at risk. However, the effect of a single 200 mg/kg dose of prophylactic dextrose gel in this trial (5% absolute risk reduction, 12% relative risk reduction) was smaller than in our previous study (18% absolute risk reduction, 32% relative risk reduction) [8]. It is common for larger multicenter trials to report smaller effects than smaller early single-center trials [11], and in this case this may relate to site variation in management of babies

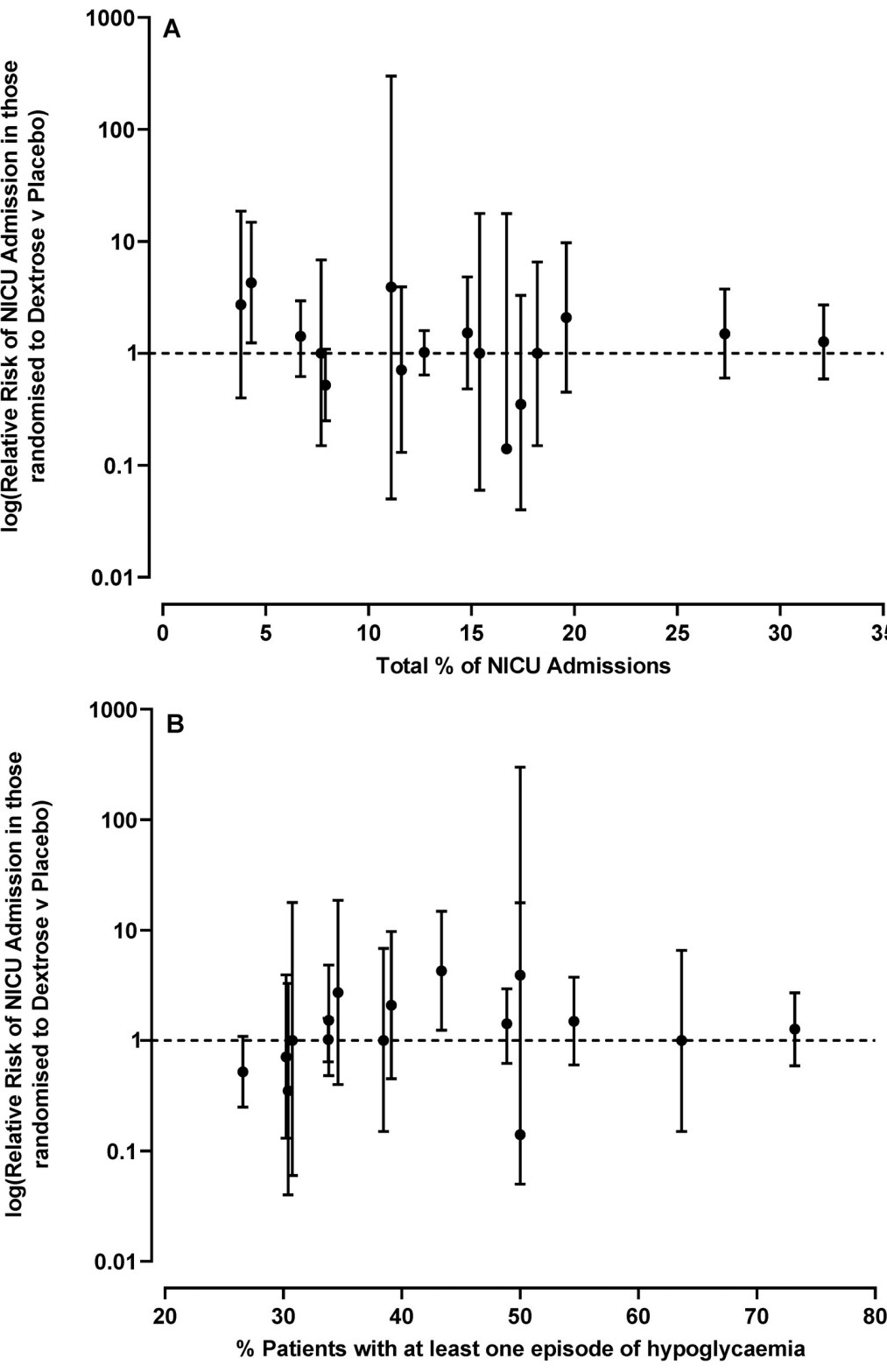

**Fig 3. Relationship between the effect of dextrose gel on rate of neonatal intensive care unit (NICU) admission and rate of hypoglycemia in different study sites.** (A) NICU admission; (B) hypoglycemia. Two study sites are excluded due to small numbers of babies recruited.

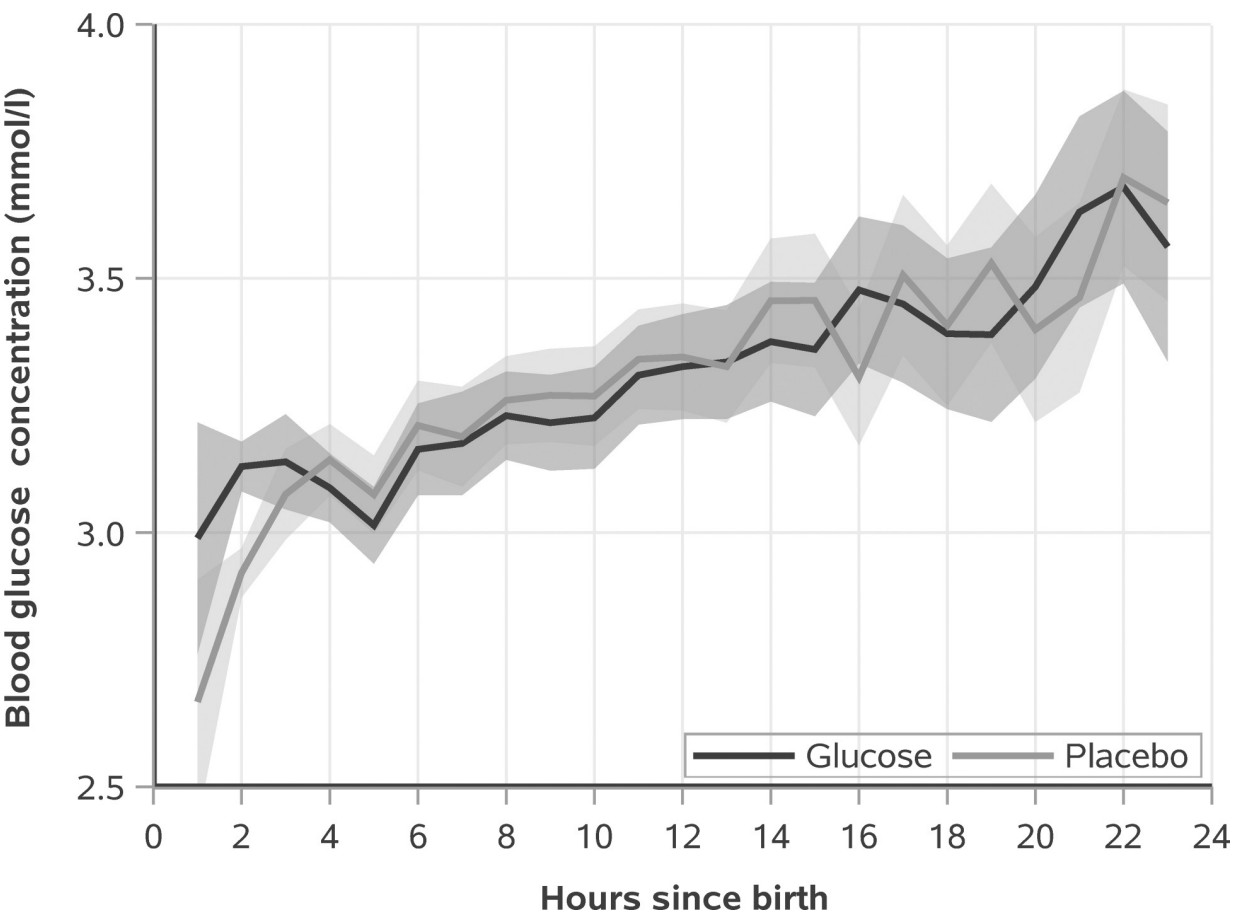

**Fig 4. Blood glucose concentrations (mean and 95% CI) over the first day for the dextrose gel and placebo groups.** Time is rounded into hour bins. Glucose concentrations are different between groups at 2 hours ($p < 0.01$).

at risk (e.g., use of formula) as well as in the detection of hypoglycemia, which was done using only accurate glucose oxidase methods in the previous study.

This trial also confirms our previous findings that a single dose of 200 mg/kg dextrose gel does not cause adverse effects, is well tolerated by babies, and is acceptable to families [8]. This is reassuring for an intervention being considered for prophylactic use in otherwise well babies, and consistent with previous reports on the use of dextrose gel for treatment of hypoglycemia [3,12].

It is not clear why prophylactic dextrose gel appeared to reduce the incidence of hypoglycemia in babies born vaginally but not in those born by cesarean section. Initial blood glucose concentrations did not differ with mode of birth in babies randomized to placebo gel, suggesting that mode of birth did not in itself alter early blood glucose regulation. Although this was a prespecified subgroup analysis, the data should be interpreted with caution in view of the multiple comparisons undertaken.

Strengths of this study include that it was a large, pragmatic, multicenter, placebo-controlled randomized trial that was adequately powered to detect a clinically important effect on the primary outcome of NICU admission. However, the majority of participants were infants of mothers with diabetes, and this may limit generalizability to other groups of infants at risk of hypoglycemia.

Another possible limitation is that some families appear to have become aware of group allocation, as more parents whose babies were randomized to dextrose gel correctly guessed the contents of the gel, and would participate in a similar study in the future. Since the dextrose gel tastes sweet, parents may have identified the gel by tasting it directly or on their babies, e.g., by kissing them. However, all study staff remained blinded to treatment allocation, and there is no reason to think that detection of primary and secondary outcomes would be likely to be differentially affected by parents' beliefs about their baby's treatment group allocation.

We conclude that a single dose of 200 mg/kg prophylactic dextrose gel does not reduce NICU admission in babies at risk. However, it does reduce the incidence of hypoglycemia, with a number needed to treat of 21 (95% confidence interval 11 to 141). Since prophylaxis also appears to be safe and is likely to be cost-effective [13], clinicians and clinical guideline groups should consider whether introduction into clinical practice is warranted at this time. The key reason for screening and treatment of neonatal hypoglycemia is to prevent brain injury, and our preliminary data suggest that use of dextrose gel to prevent hypoglycemia may improve some aspects of development at 2 years of age [14]. Later follow-up of participants in this much larger randomized cohort will be important to further assess the clinical utility of prophylactic dextrose gel in prevention of neonatal hypoglycemia.

## Supporting information

**S1 CONSORT Checklist.**
(DOCX)

## Acknowledgments

The authors acknowledge the generosity of all families who participated in the trial.

**hPOD Study Group.** Coordinating committee: Jane Harding (chair), Jane Alsweiler, Caroline Crowther, Richard Edlin, Gregory Gamble, and Joanne Hegarty. Data monitoring committee: Frank Bloomfield (chair), Katie Groom, and Thomas Lumley (all University of Auckland). Safety monitoring committee: Carl Kurshel (chair, Royal Women's Hospital, Victoria), Malcolm Batten (National Women's Hospital, Auckland), and Lindsay Mildenhall (Kidz First Neonatal Care, Auckland). Data management team: Coila Bevan, Jessica Broshnahan, Ellen Campbell, Kelly Fredell, Karen Frost, Khan Safayet Hossin, Rashedul Hassan, Grace McKnight, Robyn May, Sarah Philipsen, and Jess Wilson. New Zealand study sites: Auckland City Hospital, Auckland: Jane Alsweiler, Celia Grigg, Jodi Guthrie-Mart, Joanne Hegarty, Sabine Hulh, Andrew Meisner, Carla Saunders, and Robyn Wilkinson. Hawkes Bay Fallen Soldier's Memorial Hospital, Hastings: Oliver Grupp and Melissa Spooner. North Shore Hospital, Auckland: Dianne Allan, Susan Law, Maree Young, Jutta van den Boom, and Stephanie Williams. Southland Hospital, Invercargill: Paul Tomlinson. Tauranga Hospital, Tauranga: Karina Craine and Marian Wordsworth. Waikato Hospital, Hamilton: Alana Cumberpatch, Deborah Harris, and Rachel Ladd. Waitakere Hospital, Auckland: Dianne Allan, Susan Law, Maree Young, Jutta van den Boom, and Stephanie Williams. Whakatane Hospital, Whakatane: Maggie Sadlier and John Thompson. Whangarei Hospital, Whangarei: Ransford Addo and Wendy Taylor. Australian study sites: Angliss Hospital, Melbourne: Keith Badloo, Alice Fang, Dimitria Simatos, and Elizabeth Thomas. Box Hill Hospital, Melbourne: Alice Fang and Dimitria Simatos. Mackay Base Hospital, Mackay: Marguerette Bane and Jacinta Tobin. Mater Hospital, Sydney: Tahereh Sakhaei-Ghadriri and Bithi Roy. Tamworth Rural Referral Hospital, Tamworth: Nitin Rajput, Rebecca Sharpe, Lurena Smith, and Gavin Vajda. Townsville University Hospital, Townsville: Guan Koh, Kym Krobath, and Annemarie Lawrence. University Hospital

Geelong, Melbourne: Melissa Blake, Rachael Cusworth, Laura Eastwood, and Isaac Marshall. Westmead Hospital, Sydney: Susan Heath and James Marceau. Women's and Children's Hospital, Adelaide: Pat Ashwood, Kerry Curtin, and Anuradha Kochar.

## Author Contributions

**Conceptualization:** Jane E. Harding, Caroline A. Crowther, Gregory D. Gamble, Jane M. Alsweiler.

**Data curation:** Jane E. Harding, Gregory D. Gamble.

**Formal analysis:** Gregory D. Gamble.

**Funding acquisition:** Jane E. Harding, Caroline A. Crowther, Richard P. Edlin, Gregory D. Gamble, Jane M. Alsweiler.

**Investigation:** Jane E. Harding, Joanne E. Hegarty, Caroline A. Crowther, Jane M. Alsweiler.

**Methodology:** Jane E. Harding, Joanne E. Hegarty, Caroline A. Crowther, Gregory D. Gamble, Jane M. Alsweiler.

**Project administration:** Jane E. Harding, Joanne E. Hegarty, Caroline A. Crowther, Richard P. Edlin, Jane M. Alsweiler.

**Resources:** Jane E. Harding.

**Supervision:** Jane E. Harding, Jane M. Alsweiler.

**Writing – original draft:** Jane E. Harding.

**Writing – review & editing:** Jane E. Harding, Joanne E. Hegarty, Caroline A. Crowther, Richard P. Edlin, Gregory D. Gamble, Jane M. Alsweiler.

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
