## [Editor Report · Decision Letter 0]

20 Apr 2020

Dear Dr Harding, 

Thank you for submitting your manuscript entitled "Neonatal hypoglycemia prevention with oral dextrose gel (hPOD): A multicenter, double blind, randomized controlled trial." for consideration by PLOS Medicine.

Your manuscript has now been evaluated by the PLOS Medicine editorial staff [as well as by an academic editor with relevant expertise] and I am writing to let you know that we would like to send your submission out for external peer review.

Kind regards,

Adya Misra, PhD,

Senior Editor

PLOS Medicine

---

## [Decision Letter · Decision Letter 1]

15 Jul 2020

Dear Dr. Harding,

Thank you very much for submitting your manuscript "Neonatal hypoglycemia prevention with oral dextrose gel (hPOD): A multicenter, double blind, randomized controlled trial." (PMEDICINE-D-20-01227R1) for consideration at PLOS Medicine. 

[LINK]

In light of these reviews, I am afraid that we will not be able to accept the manuscript for publication in the journal in its current form, but we would like to consider a revised version that addresses the reviewers' and editors' comments. Obviously we cannot make any decision about publication until we have seen the revised manuscript and your response, and we plan to seek re-review by one or more of the reviewers. 

We expect to receive your revised manuscript by Aug 05 2020 11:59PM. Please email us (plosmedicine@plos.org) if you have any questions or concerns.

We look forward to receiving your revised manuscript. 

Sincerely,

Emma Veitch, PhD

PLOS Medicine

On behalf of Clare Stone, PhD, Acting Chief Editor,

PLOS Medicine

plosmedicine.org

*Per the journal's usual style, please include (ideally in the last sentence of the Abstract Methods and Findings section), a brief note about any key limitation(s) of the study's methodology.

*At this stage, we ask that you include a short, non-technical Author Summary of your research to make findings accessible to a wide audience that includes both scientists and non-scientists. The Author Summary should immediately follow the Abstract in your revised manuscript. This text is subject to editorial change and should be distinct from the scientific abstract. Please see our author guidelines for more information: https://journals.plos.org/plosmedicine/s/revising-your-manuscript#loc-author-summary

*Ideally, please reformat the in-text citation callouts so these are sequential numerals in square brackets (eg, [1], [2] etc) rather than superscript numerals - if using referencing software this should be quick and easy.

Comments from the reviewers:

Reviewer #1: 

This review relates to manuscript PMEDICINE-D-20-01227R1 about the results of hPOD trial.

Overall, I found the manuscript well written and easy to follow. I only have minor comments listed below:

* Babies for whom the primary outcome was not available were assumed to have been admitted to NICU (conservative analysis), but there was no other imputation for missing data. In the results, please clarify how many babies in each arm were subject to that imputation. Could a different, potentially more extreme, imputation scenario have led to different results?

* When describing the variables used to adjust the primary analysis model (lines 130-132), please clarify which variables entered as fixed vs random effects. Please also clarify the reason for modelling the maternal unique identifier as a clustering term (presumably due to multiple babies from a single mother) together with the method used to do so (e.g. generalised estimating equations or random effect). 

* The exploratory analysis section (lines 135-139) mentions adjustments by centre and maternal unique identifier. These terms also appear to be included in the primary analysis described in the preceding paragraph. Please clarify whether adjusting for centre and adding the maternal unique identifier was pre-specified vs post-hoc and whether it was applied to all analyses vs only exploratory ones.

* The design section mentions 16 sites whereas the results section starts by mentioning 18 sites. Please clarify / correct.

* I note that the SAP is dated August 2019 which is after recruitment ended (May 2019). Please confirm that it was written and finalised by authors who were unaware of the trial results.

* The exploratory subgroup analyses by centre (lines 205 - 209) mentions results with "data not shown". Please note that it is the journal's policy that every result discussed should be included either in the manuscript or the online supplement. Please note that the same applies to exploratory results reported on lines 229-230.

* When presenting the results of the subgroup analyses, please consider reporting the p-value for heterogeneity (that is the significance of the interaction term between the subgroup variable and the intervention) instead of the p-value testing the treatment effect within each subgroup. Please also consider using a forest plot in place of a table to present (some of) the subgroup analyses.

* Please clarify the hypothesis being tested by the analysis presented in Figure 2 (relationship between the effect of dextrose gel on NICU admission and a) rate of NICU admission and b) rate of hypoglycaemia).

* In the methods section, please indicate which subgroup analyses were pre-specified (i.e. planned before seeing any result by treatment arm) vs post-hoc (conducted after seeing results).

* The attached SAP mentions a longitudinal analysis of blood glucose measurements using a repeated-measure linear mixed model. I would like to see the results of this analysis included in the paper and presented using a longitudinal mean plot accompanied by the mean difference in blood glucose obtained from the mixed model.

-Laurent Billot

Reviewer #2: This study builds on the previous work of Drs. Harding and colleagues. Neonatal hypoglycemia remains a significant and common health problem in the neonate and there is considerable controversy regarding treatment and neurological/developmental outcomes. In their previous landmark study this group showed that neonatal hypoglycemia in at risk populations was far more prevalent than previously appreciated and that even relatively mild hypoglycemia was associated with adverse outcomes in childhood. Dextrose gel is now widely used as a treatment for hypoglycemia but it remains unknown whether or not prophylactic treatment will prevent the development of hypoglycemia and whether that matters. Thus, this current study is a critical first step to address these questions. There are a few minor issues that should be addressed as outlined below.

1. It would be helpful to the general reader to define neonatal hypoglycemia

2. I am not sure it is useful to include the mean birthweight of the entire group (line 149) as BW were different between the groups (LGA, SGA IDM, etc). It would be more helpful to this reviewer to instead have the average weights of each group.

3. Were there any differences between centers in numbers of babies in each group (SGA, LGA, etc)? I am not certain as to whether these are important data to include, but given the differences in admission to NICU between centers, it might be helpful to include.

Reviewer #3: This is a very well designed double blind randomized controlled clinical trial. The study design, methods and results are superbly presented.

The aim was to assess whether prophylactic dextrose gel given to babies at risk of neonatal hypoglycemia (NH) would reduce admission to an NICU. The investigators have clearly accomplished their aims.

My comments are all rather minor.

Blood glucose concentrations were measured at 2 hours of age (i.e., 1 hour after receiving dextrose) and thereafter according to hospital standard practice for monitoring babies at risk of hypoglycemia.

I presume the policies were not identical at all 16 participating institutions. The more frequently one measures blood glucose ("how frequently it is sought"), the greater the chance hypoglycemia will be detected. I wonder to what extent any such differences may have influenced the frequency of detection of biochemical hypoglycemia. This is discussed on P16 as a limitation of the "pragmatic multicenter trial".

Comments 

P4 line 83 a minor point: the diabetes community is on a campaign to stop referring to people with diabetes as "diabetic"

Infant of a mother with diabetes

By far the most common reason for risk of hypoglycemia was maternal diabetes (type 1, type 2 and gestational diabetes). The authors comment on this in the discussion (on P. 17).

Neonatal hypoglycemia in these infants is thought to be caused by dysregulated insulin secretion by beta cells exposed to hyperglycemia in utero. The cellular processes that eventually lead to normal beta cell function after delivery presumably take time (several hours or days) to become normal. For this reason, I am not surprised that a single oral dose of glucose per se had only a modest effect to prevent subsequent hypoglycemia caused by dysregulated insulin secretion. Indeed, oral glucose might be expected to induce insulin secretion and, in the absence of adequate enteral nutrition (carbohydrate, protein and fat) might lead to so called reactive hypoglycemia? 

Likewise, although the data are less robust, it is thought that dysregulated insulin secretion may play a role in the hypoglycemia of SGA and LGA babies.

[LINK]

---

## [Decision Letter · Decision Letter 2]

15 Sep 2020

Dear Dr. Harding,

Thank you very much for re-submitting your manuscript "Neonatal hypoglycemia prevention with oral dextrose gel (hPOD): A multicenter, double blind, randomized controlled trial." (PMEDICINE-D-20-01227R2) for review by PLOS Medicine.

I have discussed the paper with my colleagues and the academic editor and it was also seen again by 3 reviewers. I am pleased to say that provided the remaining editorial and production issues are dealt with we are planning to accept the paper for publication in the journal.

[LINK]

We look forward to receiving the revised manuscript by Sep 22 2020 11:59PM. 

Sincerely,

Artur Arikainen

Associate Editor

PLOS Medicine

plosmedicine.org

Requests from Editors:

1. Title: Please amend to: “Evaluation of oral dextrose gel for prevention of neonatal hypoglycaemia (hPOD): A multicenter, double blind, randomized controlled trial”

2. Abstract:

a. Line 29: Please clarify that “double-blinded”, and specify who was blinded to the intervention and control. Please name the participating countries here, the recruitment dates, and the number of centres.

b. Line 32: Please give the dose/concentration of dextrose.

c. Around line 32, please mention that the trial was powered to detect a 4% absolute reduction in the primary endpoint measure.

d. Line 32: Please name the secondary outcomes, and include results below.

e. Line 34: Please give child and mother demographics (% child sex, mother’s age).

f. Lines 39-40: Please provide aRR, 95%CI and p values for the following: “There were no differences between randomization groups in breastfeeding at discharge from hospital, receipt of formula before discharge, or formula feeding at 6 weeks, and no hyperglycemia.”

g. Lines 41-412: Please quantify this result: “Maternal satisfaction was high...”

h. Please mention deaths during the study and whether they were attributed to treatment.

i. Lines 42-43: Please start the limitations sentence “Limitations of this study included…”. Please also mention a second limitation.

j. Line 46: Begin with “In this placebo-controlled randomized trial, …

3. Author Summary: Please add bullet points to the start of each sentence.

4. Line 78: Missing space before citation.

5. Line 191: Figure 1 legend: Please rename to “Participant flowchart”.

6. Table 1: Recommend replacing “Prioritized Ethnicity” with “Specified Ethnicity” or similar.

7. Line 203 and throughout: For p values less than 0.001, please state “p<0.001”. Please report all p values to consistent decimal places, rather than significant figures.

8. Line 325: Recommend replacing “weakness” with “limitation”.

9. Please ensure references in the bibliography follow the Vancouver style: first six authors should be shown before “et al.”, eg for references 6, 7. Please provide a URL or DOI for reference 14. Please update the status of reference 13 from “in press” to “Epub ahead of print” or similar.

10. When completing the CONSORT checklist, please use section and paragraph numbers, rather than page numbers.

------------

Comments from Reviewers:

Reviewer #1: All my comments have been adequately addressed. 

For Figure 4, I would suggest keeping only Panel A.

Reviewer #2: My concerns were addressed.

Reviewer #3: The authors have satisfactorily responded to my comments.

[LINK]

---

## [Editor Report · Decision Letter 3]

22 Dec 2020

Dear Dr. Harding,

I am writing concerning your manuscript submitted to PLOS Medicine, entitled “Evaluation of oral dextrose gel for prevention of neonatal hypoglycemia (hPOD): A multicenter, double blind, randomized controlled trial..”

We have now completed our final technical checks and have approved your submission for publication. You will shortly receive a letter of formal acceptance from the editor.

Kind regards,

PLOS Medicine